# Did Forestland Restitution Facilitate Institutional Amnesia? Some Evidence from Romanian Forest Policy

**Marian Drăgoi** [1,*] and **Veronica Toza** [2]

1   University Stefan cel Mare of Suceava, Faculty of Forestry, 720229 Suceava, Romania
2   Green Advisers Ltd., 030018 Bucuresti, Romania; veronica.toza@greenadvisers.eu
*   Correspondence: marian.dragoi@usm.ro

**Abstract:** This paper shows how the slow process of forestland restitution, which is unfolding in Romania since 1991 has eroded the threads of sustainable forest management by an insidious institutional amnesia (IA). The four symptoms of this harmful process (frequent reorganization, transition from paperwork to electronic media, fewer people motivated to join public services, and popularity of radical changes) were analyzed from the legal standing point as well as from practitioners' perspective. After having described the legal process and the relative dependencies between laws and government ordinances we also showed that the three laws on forestland restoration (three fully operational laws and two bills submitted in 2019, one year before general elections) were produced by unintended policy arrangements. The legal loopholes of forestland restitution were described in details as well as the challenges brought about by nature conservation policy (Natura 2000 management plans *v* traditional forest planning), and the overwhelming bureaucratic burden developed to deter illegal logging, instead of fully implementing a modern system of forest watching based on volunteering. However, the main cause of IA is institutional unsteadiness which was inherited from the communist regime, and cannot be alleviated unless more involvement of professional foresters in politics.

**Keywords:** land restitution; Romanian legal system; forest policy

## 1. Introduction

The land restitution to the rightful owners who had had it before the communist nationalization was an important aim of the democratic regimes established in ex-communist countries after 1989 [1]. Since then, most of the policy makers embraced the neoliberal vision weighing and even exaggerating the role of private property and free markets; hence different privatization processes were considered supportive means for making the markets more efficient [2], abating market failures [3], or attracting new investors [4].

Moreover, two other causes of ownership restoration were also reported in literature: The emotional bonds with the ancestors or family's homeland [5,6] or, in the case of forests, the commitment to fight against corruption [7]. This latter issue is important for environmentalists because, on one hand, deforestation goes along with corruption but, on the other hand, corruption is more or less linked to the political competition as well, i.e., elective cycles [8].

Land-use change, or even land cover change, triggered by privatizing the forests is non-linear and typifies many transition economies [3,9,10]. However, due to important ecological, social and economic outcomes, this process may also generate a negative social feedback able to reduce the deforestation rate if the external socio-economic aspects facilitate reforestation works, as reported in Vietnam [11].

After 1990 the Romanian forest policy has been undermined by a three-step process of forestland restitution featuring a single important stakeholder: The National Forest Administration (NFA), who has steered the process of restitution and has been managing most of the protected areas ever since [12,13]. This important role played by an entity considered *de facto* the real landowner of the public forests has fed social mistrust in NFA, sometimes justified, sometimes not: Villain *or hero*? This question still lingers on the NFA public image, for many reasons, including the unfair competition with private forest districts [14].

Even though a great deal of forestry literature lately published abroad dealt with illegal fellings and deforestation occurred in Romania [14–17], little has been published about the unstable institutional framework created during the restitution process [18–20]. Due to forestland restitution, the Romanian forestry moved from a heavily centralized sector to a multilayer type of governance [21], with more actors, playing different and even conflicting roles, as further explained in the results section.

Regarding the forest policy after 1990, the most paradoxical situation is that, despite the rigorous legal framework, subsidies for watching services, and expensive institutional settings, Romania became famous for illegal fellings. Corruption is one explanation [14], but it cannot be the only one, and this study tried to explain in detail a series of hindrances of Romanian forest policy, so far overlooked by the mainstream scholars.

Forestry, biodiversity conservation and agriculture are inevitably interconnected in what Christopher Pollitt regarded as public service networks [22]. As none of these networks relies on a single organization, be it a public authority, or an association or whatever type of juridical entity, inevitably these networks change over time, for different reasons. Ideally, each public authority should deal with only one network, not with more or less isolated sectoral economies. Policy changes are being steered by these networks wherein no single organization is able to preserve and completely retrieve (when necessary) the memory of one or more processes unfolded in the past [23].

The provocative topic of institutional memory popped up on the scientific agenda by late of 80′ being interconnected with the problem of bottleneck of eliciting knowledge from experts to produce the first generation of expert systems [24]. Institutional memory consists of those parts of organizational memory independent of any member of the organization: "if swapping two members of the organization" doesn't alter the organizational memory, that memory is purely institutional [25].

Romanian public authorities responsible for developing and implementing forest policy are typical post-bureaucratic organizations because they are no longer based on stiff hierarchies, appointments are made on ever-changing criteria, more or less transparent, salaries are less uniform and less predictable, and a lot of work is outsourced, or is carried out on part-time bases [22].

By this study, we aimed to evaluate the extent to which the inconsistent forest policy brought about institutional amnesia (IA) of the forestry department since 1990, when the communist structures of governance have started being dissembled.

The rest of paper is organized as follows: After having defined the four symptoms of IA, we identified clear-cut evidence of IA, presented in the results' section. Some suggestions regarding further progress to be made in the organizational culture of forest department are presented in the discussion section, and finally some conclusions are drawn in the last section.

## 2. Materials and Methods

### 2.1. Theoretical Premises

Quite often politicians make more politics and less policies: In other words, quite often the laws fail to address the needs that society must face; yet, luckily, the professionals employed by the public authorities are the ones that fine-tune the laws in order to put them in force. Peculiar to the forest policy, one plausible explanation of the gap between what had been intended and what has been effectively achieved by implementing forest-related laws shall be sought in the fact that institutional settings are being made (and inherited) by professionals, while the laws are being made by politicians,

most of them having quite narrow competences in forestry and related technical realms, like cadaster, forest inventory or forest planning.

The extent to which the laws are socially acceptable may vary from law to law [26,27]. Unfortunately, the issue of social acceptability is completely overlooked by Romanian policy makers, and the only study produced in this respect was focused on short-rotation poplar plantations [28], which are not actually addressed by the Romanian legal system in place.

Therefore, a thorough analysis of the difficulties faced by the forest owners, professional foresters, and supervising authorities is important for understanding why the forest legislation has long failed in addressing issues like illegal logging, conflicts among stakeholders, timber market failures, and so forth.

We have tested whether, and if so, to which extent, the four symptoms of IA, defined by Christopher Pollitt [29] hold for Romanian forest policy in the last three decades. Diagnosing the IA is important because it goes hand in hand with the drift to low performance, as defined by [30]. For the sake of consistency, we cited the first words of the symptoms as defined by Pollitt in italics, within brackets. The four symptoms of IA were rendered into the next four hypotheses to be tested:

1. (*Increasing rate of organizational re-structuring*) Frequent organizational changes of authority brought about personnel instability and fewer reliable forest-related data the authority may count on.

2. (*Rapid shifts in the media in which records are held*) Transition from paper-based information flows to electronic media did not reduce bureaucracy; on the contrary, too much freedom in changing reporting templates and forms overwhelmed the executive staff of NFA and Forest Guards (FG) at the expense of data accuracy, significance, and relevance. Having less data from the past, knowledge loss is inevitable.

3. (*Decline of the concept of public service*) Fewer graduates are committed to a permanent career in forestry public service; thus, the new organizations established meanwhile are deprived of their own institutional memory (like happened with FG).

4. (*Popularity of ideas of unceasing, radical change*) Public authority, private forest owners, logging companies and environmental activists put pressure on NFA to get reorganized at any cost ignoring that NFA has been the only public institution able to manage the public forests and handle clearly the restitution process. On the other hand, NFA has mimicked internal reform by externalizing most of the forest operations, which also contributed to IA.

### 2.2. Policy Arrangement Approach

The methodological framework wherein this study falls better is *policy arrangement approach* (PAA), based on four interconnected factors: Political coalitions, rules of the game, discourses and power relations [31]. In our case, the rules consist of the whole mechanism of land restitution (the local and county commissions, the procedures to follow), discourses are the political slogans conveyed to voters while the power relations refer to the ad-hoc networks of interests created during political negotiation in Parliamentary commissions of agriculture, forestry, services, and food industry. The video records of these debates are available on the two chambers' websites since 2006 [32,33]. The slogans and the political coalitions that promoted different bills on forest restitution are shown in Table 1 while the rest of the information was inserted as commentaries.

### 2.3. Desktop Research

Since the four hypotheses aforementioned have to be tested and documented, we designed two desktop studies, five semi-structured interviews and one inquiry on the agenda of the public authority in charge with implementing forest policy, as reflected by the subjects debated by the Technical Council for Forestry (TCF) pending to Forestry Department, which is the collective decision-making organism of the public authority.

**Table 1.** Snapshot on the four laws of forestland restitution.

| Law | Key Institutions | Area Threshold Per Claim (Hectares) | Most Threatening Provisions, Misinterpretations or Omissions | Institutional Changes; New Stakeholders; Policy Arrangements (Coalition, Discourse) |
|---|---|---|---|---|
| 18/1991 | LCLR, CCLR, NFA | 1 ha forest, 10 ha agricultural land | " … preferably, the land will be restituted on the location specified by the claim" No penalties for LCLRs disobeying or misinterpretation of the legal provisions [34]. | No institutional changes, no policy arrangement |
| 1/2000 (known as "legea Lupu") | FG, NFA, CCLR | 10 ha forest, 30 ha agricultural land per individual claim. No threshold for joint ownership, 30 ha for juridical persons | Two eyewitnesses' testimonies to support any claim Forest roads not restituted but the land beneath them restituted | FG, Private forest districts, National associations of forest owners; Coalition: Democratic Convention issued the bill before local and parliamentary elections; Discourse: *"NFA is deeply corrupted and robs our parents' woodlands"* |
| 247/2005 (initiated in 2004) | NFA | No limit | In Transylvania, after the Great Unification of 1918: Forests bought back by the Romanian state from Hungarian owners were restituted after 2005 as if they were confiscated by the communist regime. Many forests located in protected areas were restituted. | Middle-men buying litigious ownership rights from individual claimants Large companies able to buy all logs illegally harvested, more protected areas (Natura 2000 sites); Ruling coalition: Justice and Truth (all right-wing parties); by the end of 2004, Discourse: *Land restitution won't ever be a political enticement for more votes"*. |
| 165/2013 | National Agency of Cadaster and Real Estate Advertising, NFA | No limit | Brand-new local commission to inventory the available land for restitution Precise locations of available land precise deadlines for finalizing the restitution process | European Court of Human Rights (ECHR), Natura 2000 network, Foreign citizens and companies, investment funds. No policy arrangement, law was required by ECHR. |

The first desktop study was focused on forest-related laws and regulations, and their side effects. The laws considered relevant to the purpose of this article refer to (1) forest restitution, (2) forest management in a broader sense, (3) Natura 2000 network, and (4) institutional settings. The legal context was reframed and updated using the information provided by the portals of the Ministry of Justice [35], Parliament Representatives' Chamber [32], Senate [33] and the Official Journal of Romania, where all normative acts i.e., Laws, Ordinances (GO), Emergency Ordinances (EO), government decisions (GD), and ministerial orders (MO) shall be published before entering into force. The dependencies between these various normative acts are outlined in Figure 1, where the arrows indicate the subordination and co-ordination relationship between different levels of jurisprudence.

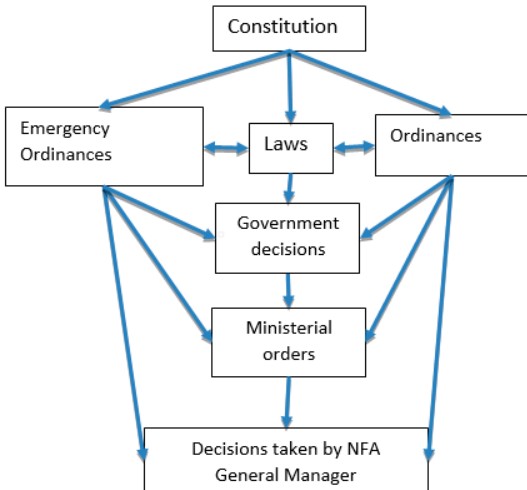

**Figure 1.** Legal process and internal dependencies.

Afterwards we checked if we figure out the effective consequences and side effects of the legal system; five semi-structured interviews were carried out to check whether or not the most relevant pieces of legislation were addressed, and if the ancillary consequences were analyzed. The interviewees stood for the stakeholders involved in or affected by implementing the legal framework: The public authority (VL, and DP former secretaries of state), Forest Guard (BM, chief inspector), NFA (SG, head of NFA county branch), and private owners (LG, the beneficiary of 600 hectares of forest). We opted for a big private forest owner who underwent all juridical stages of forest restitution.

A second desktop study was based on the information provided by the portal of Tribunals and Courts [36], where, using some keywords, we found all previous and ongoing lawsuits related to restitution process, which are their terms, the causes and the solutions, as well as the plaintiffs and claimants. Each court's website was inquired with the following key-words (in Romanian language): "Romsilva" (NFA is denomination in English of the National Forest Administration), and "fond funciar" (land title). Then all significant records were copied and tagged into a spreadsheet where the trials have been dated, located and tracked down at different judicial levels (ordinary court, tribunals, courts of appeals, High Court of Cassation and Justice). The time window covered for this desktop study was 2005-2018, although few cases were older; all in all, 2514 cases were found. However, these cases do not cover all relevant forest-related casuistry because the juridical disputes have resumed in 2018 when NFA sued the Orthodox Church for misreading a provision of Law 1/1991. Yet, this series of trials gave a glimpse on the ever-lasting potential conflicts between the state and the private forest owners. Given the unique ID of each case, it was possible to keep track of each case from regular trial (one single judge decides) to 1st appeal (two judges decide), 2nd appeal (three judges decide) and action in annulment, when the decision is made by the High Court of Cassation and Justice.

A third desktop study was carried out on the agendas of TCF since 2017. This commission approves not only forest management plans, but also all derogations regarding the allowable cut,

irrespective the reason. This study was meant to document the symptoms of IA related to forest planning and timber sales.

## 3. Results

### 3.1. Unfolding the Legal Process

The legal process can be a primordial cause of IA and therefore it must be shortly explained before going further with presenting its results. Even though some lapidary regulations on legislative technique have been inherited from the communist period, the Romanian Parliament has promoted a standard procedure on drafting normative acts only in 2000 (Law 24/2000). Prior to that, the legislative technique was regulated only by the Romanian Constitution, adopted in 1991, and a presidential decree dated back in 1976.

Figure 1 depicts the existing legislative hierarchical mechanism relevant to forestry sector. Customary, a law is initiated by the Parliament, or by 100,000 citizens at least, under some particular circumstances. Another way, quite often used, is the so-called legislative delegation, meaning that a law is being initiated by the Government as GO, except the domains regulated by organic laws (like forestry).

The laws fall into two large categories: Ordinary laws, passed if voted by a simple majority of the present members of both chambers, and organic laws, requiring the majority of the total members of each chamber (Romanian Constitution, art. 5).

The Forest Code is an organic law and it cannot be amended by ordinary laws; this explains why the Forest Code was amended so often in the last two legislative cycles, (2012–2020) ruled by a coalition of two parties, and barely amended before: Being an organic law, the slightest amendment requires a large majority, and that majority was hard to come by before 2012.

There are two different situations when GO can be adopted: (1) During the Parliament vacation, the Government is being enabled to issue regular GO, except the ones required for strategic domains; (2) under very specific circumstances, the Government is entitled to adopt even Emergency Governmental Ordinances (EGO). Irrespective of these differences, GOs and EGOs shall be subsequently approved by the Parliament as laws, mandatorily. However, the Romanian Government has long started governing through EGOs, this issue being subject to permanent political disputes raised by the opposition parties, civil society, and other stakeholders, including the media.

However, the Parliament internal regulations do not provide for any timeframe or deadline for finalizing a new law. Hence, every new draft, regardless of its initiator (be it a member of the Parliament or a ministry) could lounge for years between the two chambers or even before being submitted by the Government (as actually happened to the Forest Code between 2005 and 2007). Whenever political interests have prevailed, the same Forest Code has been amended within months, as happened in 2013, 2015, 2017, and 2018. Many amendments were required by The Constitutional Court, meaning that old articles were not constitutional at all.

### 3.2. IA Symptoms

#### 3.2.1. Forestland Restitution—A Political Abiding Process (All Symptoms)

The relevant stakeholders and institutions involved in forestland restitution are depicted in Table 1, while the most important loopholes of the four laws of forestland restitution are described in Table 2. All conflicts between the State and claimants have been caused by the way in which the forestland was effectively transferred from NFA to each private owner. All claims approved in a CCLR session were centralized in a database and sent back to LCLRs and NFA county subsidiaries. Having approved the ownership transfers, NFA had effectively transferred the land to LCLR, supposed to finalize the process. About 500,000 hectares of forests were left in LCLRs custody, (without any watching services), because the claimants had refused the ownership titles for not being located precisely in the place

where their parents had had those forests. As LCLRs had no legal responsibility in forest watching, all these forests were presumably illegally logged [37].

**Table 2.** Key stakeholders of forestland restitution.

| Stakeholder/Institution | Roles Played in Restitution Process | Legal Basis |
| --- | --- | --- |
| Local Commissions of Land Restitution (LCLR), County Commission of Land Restitution (CCLR) | Gathers the claims from LCRL and checks the claims for geographical consistency within each county considering the available forestland and the differences between administrative precincts | Law 18/1991 |
| Forest Guard (FG) | Territorial structure in charge with implementing the forest legislation (forest regime) and checking the timber flows (from stumpage to sawmills and further on). Different names given under different public authorities | EO 169/1998 |
| NFA | The holder of the public forests in lieu of the Ministry of Finance | Romanian Constitution |

In 2000, not surprisingly for the political commitment of the rightwing coalition that ruled between 1996 and 2000, the restitution process was over-simplified by replacing any solid evidence (i.e., legal documents) that could attest the ownership over the property with two witnesses' testimonies (Law 1/2000). In two historical provinces (Wallachia and Moldavia) lacking cadastral books [34], this new law opened the door for many counterfeited scams, followed by numerous trials.

The political determination was so high that even "*a vague lawfulness*" was enough for kicking off the restitution process, as the interviewed FG chief inspector said. The total area of private forestland sharply increased, and the first private forest districts have been created, most of them subordinated to municipalities [38].

For the stakes of the final stage of restitution (Law 247/2005) were much higher (less available land, inevitably) the claimants had to produce solid evidence for their requests. Perfect time for a special type of stakeholder to show up on the scene: Middlemen - rich businessmen, with a good grasp on the juridical system, bold, tenacious and able to produce, if needed, fake documents, resembling the ownership titles issued seven decades ago, yellowed by time ("a *microwave oven and some experience in cooking the paper were enough*", confessed the interviewed private owner). The same forest owner stated that all three laws of land restitution were useless: "*the whole process would have been more cost-effective if each claimant had brought the Romanian State before the court, invoking the Constitution which says that private property is protected and guaranteed by the State*".

The current distribution of forest against ownership types (public *v* private) is presented in Figure 2.

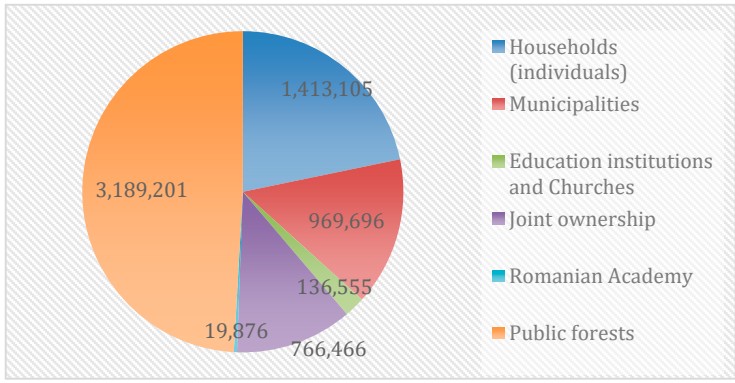

**Figure 2.** Forest Area against ownership types Source: 2017 Annual Report on the Forest Condition, Ministry of Forests and Waters [39] Numbers represent areas, in hectares.

The social dimension of potential conflicts between the private forest owners and the pubic authority is better rendered by the Pareto graph from Figure 3, which shows that properties smaller than 10 ha stand for close to 99% of the individual owners, who possess about 40% of the private woodland. Forest sustainability on less than 10 hectares is even more elusive if someone would ever try to convince 346,015 individuals to pay for a management plan whose technical provisions befit large public forests [40,41].

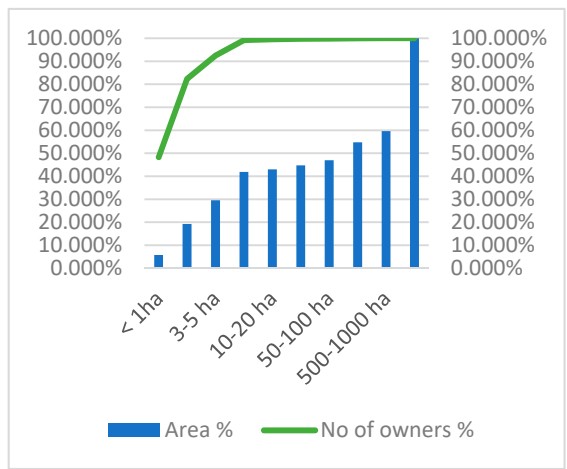

**Figure 3.** Pareto histogram of individual forest ownership.

In the first half of 2019, prior to EU Parliament and presidential elections, the same ad-hoc coalition occurred again: The ruling coalition (made of two parties) came with a bill meant to resume the restitution process [32], the target groups being the churches and some historical communities unable to testify their land tenure when the second wave of restitution had started, in 2000.

Mirroring in detail what had happened in 2004, the rightwing opposition came up with a simple and seemingly harmless amendment to the Forest Code, which adds to the definition of the public forest the effective forest area owned by the state prior to communist nationalization (i.e., Law 177/1947), which was just 1,942,000 hectares. The difference to the actual area of the public forests (more than 3 million hectares, see Figure 2) shall be restituted to other entities (municipalities, joint ownership, and companies) whose successors failed to produce solid tenure evidence so far, for whatever reason. Among the motives invoked by this new bill is Law 165/2013, art. 13, 1st paragraph, which states that if the land restitution is no longer possible on the initial locations, the corresponding area will be restituted from the land owned by the administrative precinct where the claim were submitted. Other reasons are the new Civil Code, which better defines the public ownership and the decision made by the Constitutional Court in 2017 [42].

Recalling the previous laws on land restitution (issued by the end of elective cycles), and the power relations between main parties, a new policy arrangement is obvious.

After 2004, NFA tried as much as possible to delay or stop the restitution process by litigating each and every single important claimant whose submission had been approved by the CCLR.

Figure 4 shows the dynamics of the lawsuits between NFA and private owners against the four levels of national jurisdiction. Most of the trials occurred after the third wave of land restitution, being initiated by NFA mainly.

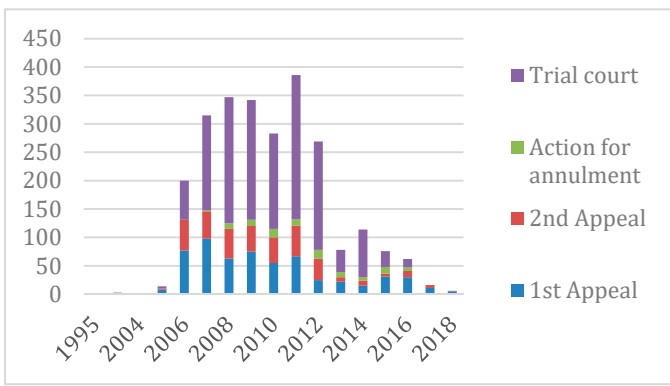

**Figure 4.** Trials national forest administration (NFA) *v* forest owners.

### 3.2.2. The Intricate Evolution of the Restitution Process Molded by International Jurisdiction (All Symptoms)

The legal trials' toll on restitution has been taken at international level, only the ECHR having recorded several thousand legal trials related to restitution process and property protection in Romania. As many as 1229 solutions have been pronounced only in relation to the Protocol no.1 to the Convention, Article 1.1—*Protection of property*, which is directly related to the right to property and its legal protection [43].

Nevertheless, given the high number of restitution cases that ECHR was entrusted to solve, ECHR issued in 2010 a pilot judgment, in the case *Maria Atanasiu and others v Romania*, in which ECHR singled out the deficiencies of the restitution process mechanism, indicating to the respondent State (i.e., Romania) new steps needed to be taken in order to process the restitution claims with higher efficiency [43]. *Inter alia*, Romania was summoned to adopt the necessary measures to finalize the restitution process within 18 months since the judgment; meanwhile ECHR would suspend all cases originating from Romania on this specific topic.

Consequently, Romania promoted a new Law, no. 165/2013, meant to close down the restitution process. Or, at the least, this was sought to achieve. Yet another pilot judgment pronounced by ECHR in case *Preda and others v Romania*, ascertained in 2014 that most of the necessary measures and legal remedies have been taken without covering all restitution-related situation. However, the restitution process and its legal battles are still going on.

### 3.2.3. Blockage of Timber Market (1st Symptom)

The public scandals on illegal fellings, triggered and fed by the media, were confirmed by the Romanian Court of Accounts, who published an extensive report on all major illegalities produced on (and with) forestland after the year of 2000 [36]. The public message conveyed by this report was set on illegal fellings and the Parliament come up with a long list of amendments to the Forest Code afterwards. Among the new provisions there is a special regulation on wood market, and the first one (pending to NFA general director) was adopted by the end of 2015. Since then three new regulations came into force, each one worse than the previous one. Initially NFA could sell only wood on the stump having the average reserved price equal to the average price of the previous year, thus triggering an unsustainable price spiral [44].

The next year's regulation has corrected the price problem, but NFA couldn't harvest by its own no more than 20% of the allowable cut, the rest of the volume being sold on the stump. In 2018 the regulation went to the opposite direction, in the sense that NFA had to outsource all harvesting operations, and sell the logs at the roadside or from log yards, despite the poor logistics and infrastructure [45].

Changing so often the harvesting regulation overwhelmed the NFA staff and the field works have been delegated to inferior ranks, like technicians and even forest rangers. If timber cruising and

post-harvesting assessment procedures are carried out by the same staff, there is still much room for corruption, even though the illegal fellings almost disappeared from the public agenda.

The timber market was disturbed by another law, justified by the numerous lawsuits triggered during the restitution process: According to Law 374/2006 all harvesting operations, including sanitation cuttings, are completely prohibited in forests whose ownership is still debated in courts. The magnitude of this side effect is unknown because the Ministry of Waters and Forests has no official records on this issue.

### 3.2.4. Forest Management Planning (2nd and 3rd Symptoms)

After forests' nationalization by the end of 1947 all Romanian forests have been thoroughly described, mapped and organized into production units (PU), each PU having its own allowable cut and cutting budget for the next ten years. A modern functional zoning system has been implemented since 1954 [46] to better match the ecosystem services provided by forests. It was all about longer rotation and silvicultural systems based on natural regeneration.

The forest planning system had been implemented by one of the most resource-demanding software run on the computer mainframes of late 70s. Nowadays, even though many forest districts have GIS maps [47], the management plans do not allow any customized GIS filters, nor the basic file-related facilities provided by any modern software, like data import and export in standard formats, such as 'mdbf', 'dbf', 'csv', or 'xls'.

Since all computer programs currently run in forest management (in a broader sense, not strictly those used for forest planning) were developed independently, without any coordination or predefined standards for input and output, none of the end-users, including the public authorities, may take advantage of a minimal interoperability of data. Consequently, the forest plan database cannot be fed or updated with fresh data retrieved from other programs, like those used for timber cruising or forest protection. Simple accounts at stand level, like the volume available on the stump, or currently thinned area, are to be carried out manually. Therefore, the extended report on the decision support systems applied in European forestry, produced by the COST action ORCHESTRA in 2018, did not mention anything about Romania despite the effort spent on finding something in common with what is currently happening at European level [48].

Even worse, each new forest plan cannot be linked to the previous one, even though the two plans refer to the same PU and the same compartments and sub-compartments. Hypnotizing that data mining algorithms were at hand, it is impossible to find out where the flowed data (systematic lower heights and/or higher ages of stands) had been reported, without professionally hacking other files, stored in different places, in different formats.

Worth mentioning, the number of forest districts did not increase at the same pace: Before forest restitution, the public forests were managed through near 400 forest districts; now, according to official statistics, there are 572 forest district [44] and 4865 logging companies [49]. In addition to its main tasks, the TCF must approve the documentation submitted by forest districts affected by windthrows and forest pests in order to reduce the current cutting budgets of those forest districts. Eventually, everting comes down to harvesting no more than the allowable cut given by forest harvesting plans, and closing all legal loop-holes used by logging companies, forest owners, and forest managers [50].

So far as many as 302 management plans of Natura 2000 sites have been approved and endorsed by the Official Journal of Romania. According to the law, the management plans of the forests located in Natura 2000 sites shall fit into the management plan of the protected area within 12 months, regardless the validity period of the forest plan, and that new forest plan shall be approved by the Biodiversity Service of the Ministry of Environment.

By the end 2017 as many as 156 new forest management plans of forests located in Natura 2000 sites have been updated according to Natura 2000 management plans, but these new forest plans were not approved by the Ministry of Environment because they were produced prior to the expiration of the old ones. According to the same Forest Code, the cutting budget needs being updated

whenever salvage fellings must be applied on large areas due to heavy disturbances like windthrows or bark-beetle attacks in Norway spruce forests. These updates are called *addenda*. However, if the cumulated harvests exceed the decennial allowable cut, any further harvesting operation is banned until the next management plan is in place, according to the same Forest Code. All these addenda shall be approved by the TCF, which is an important bureaucratic burden, often delaying other important decision to take.

Compared to the situation before 1990, when all forests were public, and the management plans were updated every year for 40 forest districts on average, in 2017 and 2018 as many as 786 new management plans and addenda to the existing ones were checked and approved by TCF.

The legal loophole is still there, in the Forest Code, where an article precisely specifies: Forest *management plans cannot be updated prior to their expiration* (that is no sooner than ten years). In other words, harvesting operations could be banned because the Forest Code was not properly amended in order to make the difference between two completely opposite situations: (1) When the decennial allowable cut was harvested prior to expiration of the forest management plan, and (2) when the forest plan shall be mandatory updated, according to the harvesting constraints brought by Natura 2000 conservation measures.

### 3.2.5. Institutional Unsteadiness (1st and 3rd Symptoms)

After 1990 the Forestry Department has been included into three different ministries, in different setting and the institutional unsteadiness is best shown by the frequent nominations of the secretary of state in charge with forestry: A new secretary of state was nominated almost every year, in the last decade [44]. As shown in Table 3 this unsteadiness has a much longer history, dating back in the communist period.

The present FG network has changed its formal name three times but more often its territorial competences. FG network has been created for watching timber cruising irrespective to the ownership, for supervising afforestation works financed by the public budget, as well as for checking and approving the forest management plans. The first legal document legitimating these territorial structures was EGO no. 96/1998, according to which every FG regional office encompasses more counties having 100,000 hectares of forest at least, while the total area controlled by one forest inspector shall not exceed 10,000 hectares.

The government decision, which endorsed the ordinance, had been barely issued two years later, in 2000 (GD 1046/2000), when six territorial FG were established. In 2003, a new Government Decision (GD no. 761/2003) was issued in order to separate the two main functions exerted by FG: Forest extension, on the one hand, and prevention of illegal logging, on the other hand. In fact, this GD created not only the FG, but also the National Guard of Environmental Protection, aiming at preventing, identifying and suing for whatever environmental crimes.

The network established in 2000 was dismantled three years later in 16 territorial Inspectorates subordinated to the National Environment Protection Guard. One year later, in April 2004, the FG network was again reorganized in just eight territorial units. This correction brought severe personnel reduction, which lessened the effectiveness of all measures meant to discourage illegal cuttings. These structures have been authorized to implement and supervise two important projects: The SAPARD program, meant to draw up money for rural development, and the Forestry Development Program, launched in 2003 and supervised by the World Bank [38].

In 2005, when the third election cycle had begun, the FG network was reorganized again (Government Decision no. 333/2005), and that structure has been maintained ever since. Each and every change of the FG network had been made only for political reasons; a former secretary of state, declared when it came to one the FGs located close to the wester boarder: " ... *that FG came up overnight, it was just politics behind*".

Barely in 2016 the wages of FG inspectors became competitive, compared to the salaries paid by the NFA. The old-fashion control of illegal logging, based on checking the transportation documents

was about to come to an end by 2016, when the ministry has launched the trial version of a smartphone application allowing online wood tracking (a screen capture is shown in Figure 5).

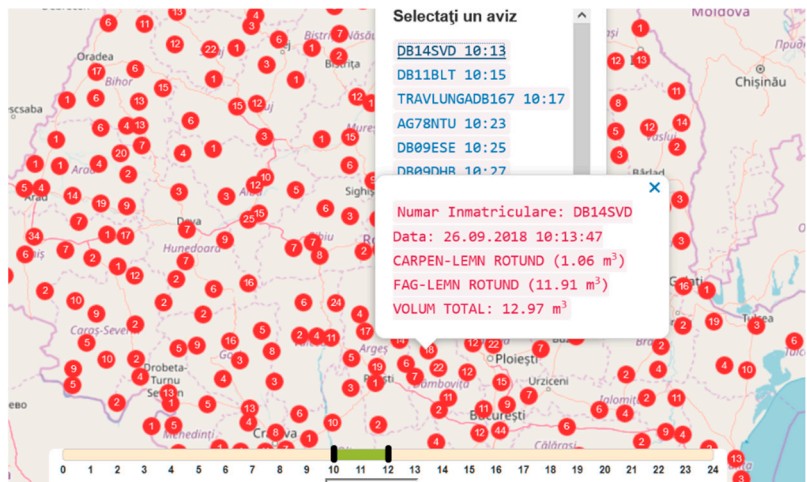

**Figure 5.** Screen capture of www.inspectorulpadurii.ro (displays the truck's and tender's license plate numbers, the species, the timber grade and the total volume transported by each vehicle (identified by its plate number) from every harvesting operation site [51].

Unfortunately, the bundle of protocols and procedures required by European Timber Regulation EUTR 995/2010 on wood tracking, including the aforementioned smartphone platform, were prorogued two times since 2016, and barely by the end of 2019 the new wood-tracking system will be operational. The formal endorsement, which is a ministerial order, was published into the Official Journal of Romania by the end of 2018. Could this unjustified prorogation convey a sort of 'human' understanding for small-scale illegal fellings? It could, of course, unless other signals have been provided by the National Forest Inventory [52], whose official site does not provide any information about the total harvest (legal and illegal fellings), neither at the end of the first cycle, nor after the second cycle.

**Table 3.** Institutional settings of Romanian forest sector since 1948.

| Period | Public Authority in Charge with Forestry |
| --- | --- |
| Before nationalization (1948) | Ministry of Agriculture and Domains |
| 1948–1949 | Ministry of Silviculture (58 regional offices, 467 forest districts). Harvesting operations and hauling were coordinated by the Ministry of Industries |
| 1950 | Ministry of Silviculture and Wood Industry (28 regional subsidiaries one for each administrative unit, 330 forest districts, in charge with all forest works, including harvesting operations) |
| 1951 | Ministry of Silviculture (same regional branches, but harvesting and hauling went to another ministry) |
| 1952 | Same as above but the regional offices were reduced from 28 to 18 |
| 1953 | Ministry of Agriculture and Forestry |
| 1956 | Ministry of Forestry (including harvesting operations) |
| 1957 | Ministry of Agriculture and Silviculture |
| 1959 | *Ministry of Forest Economy (silviculture, harvesting and wood industry)* |
| 1969 | Superior Council of Agriculture |

**Table 3.** *Cont.*

| Period | Public Authority in Charge with Forestry |
|---|---|
| 1972–1982 | *Ministry of Forest Economy and Materials for Constructions* |
| 1982–1989 | Ministry of Silviculture (harvesting operations and wood processing were coordinated by the Ministry of Materials for Constructions) |
| 1990–2000 | *Ministry of Environment, Water and Forestry (harvesting operations coordinated by the Ministry of Economy)* |
| 2001–2003 | Ministry of Agriculture, Food Industry and Forests (represented across the country by 16 Territorial Inspectorates of Forest Regime and Game Management |
| 2004–2005 | Ministry of Agriculture, Forests, Water and Environment—8 Territorial Directorates of Forest Regime and Hunting |
| 2005–2009 | Ministry of Agriculture, Forests and Rural Development (9 territorial Inspectorates for Forest Regime and Hunting) |
| 2009–2014 | Ministry of Environment, Water and Forests |
| 2014–2015 | Ministry of water and Forests (Ministry of Environment for biodiversity issues) |
| 2016 | Ministry of Environment and Climate change (Forest guards instead of forest inspectorates) |
| Since 2017 | Ministry of Waters and Forests (Ministry of environment and climate change for biodiversity issues) |

Compiled by inquiring the collection of "Revista Pădurilor", available online [53].

### 3.2.6. Harmonization of the Legal Framework (2nd and 3rd Symptoms)

Another stream of legal enforcement (Laws, GOs, EGOs or GDs) comes from the fiscal, territorial planning and environmental policies. As long as the forest policy is not fully integrated into a wider policy, like agriculture or environmental protection, the policy makers must handle simultaneously at least two flows of technicalities already in place, coming from the Ministry of Water and Forest, and from the Ministry of Environment, respectively. These two authorities are producing even conflicting norms, GDs, and ministerial orders, puzzling the low-level executive decision makers, like the general director of NFA, as the ex-chief of FG stated.

Worth noting, all contradictions between laws are solved by the Constitutional Court, but the contradictions between two or more low-level regulations, such as ministerial orders cannot be addressed by any juridical means. For instance, the Constitutional Court rejected the Law, which endorsed EO 100/2004 according to which about 90,000 hectares of forests would have been restituted to the Orthodox Church of Bucovina. Not surprisingly, that EO was issued by the very leftwing Government just before the parliamentary elections of 2004, in an attempt to outcompete the popularity gained by the political opposition thanks to the bill on land restoration (Law 247/2005), which had been promoted by the rightwing opposition parties by the end of the third elective cycle (2000–2004).

Lacking proper implementing measures, many provisions of the Forest Code (Law 46/2008) couldn't be enforced despite the goodwill and foreseen positive results: Subsidies for forest watching on forest properties smaller than 30 ha, or payments for the ecosystem services (promised by the previous Forest Code (Law 26/1996), are just two of the notorious examples that have eroded public and stakeholders' confidence in public authorities.

Even worse, having these obligations undertaken by the Romanian State through the Forest Code, some big forest owners sued the Romanian State for not obeying the legal provision of compensatory payments, not only for protected areas, but also for all private forests assumed to provide ecosystem services, although their proper monetization is still awaiting [54].

Since 2005 a private foundation (Conservation Carpathia) has been buying illegally logged and abandoned forestlands, or even badlands, for ecological restoration. Barely in 2016 their officials acknowledged the main goal to create a private National Park in Făgăraș Mountains, overlapping the

actual Natura 2000 site. If this private entity voluntarily raises the protection status from Natura 2000 standard to National Park standard, it should not ask for compensation simply because the landowner decided to change the protection status, not the State. Such a claim conveys a pure rent-seeking behavior [55] because Conservation Carpathia bought the forestland not for using the timber, but for taking advantage of a legal provision, meant to compensate the forest owners who *were deprived* by their right to harvest any tree *after* being entitled as landowners, and *not before*. Such an attempt should be regarded as nothing but a scam, which is a sort of corruption [56].

### 3.2.7. Long Political Disputes on Deceptive Subjects (4th Symptom)

In Parliament essential provisions of whatever law can be ruled out or modified through political negotiations. For instance, the draft of the latest Forest Code loitered two years (2005–2007) in the Ministry of Agriculture and Forestry, and one year between the two Parliament chambers, going back and forth for two interconnected provisions seemingly contradicting the ownership rights and the Civil Code. Both articles refer to the small forest properties only; the former article, dating back to the Forest Code of 1910, stated that a private forest cannot be split by inheritance in properties smaller than one hectare, while the latter article established an area threshold wherein a new construction can be built into a private forest, including the access route and the backyard. That threshold area, to accommodate different situations, is 5% of the total forest property, but not larger than 200 m$^2$. The EGO 193/2008 (endorsed by Law no. 193/2009), shortly repealed those two articles, meant to deter the urban sprawling [57] invoking the Civil Code provisions.

Having these two articles ruled out, the headway to urban sprawling real estates in private forests was opened. Another serious drawback of the Forest Code issued in 2008 was the haste to repeal all subsequent regulations issued after 1996, when the prior Forest Code came into force. Having neither GOs nor ministerial orders behind, the new Forest Code became gradually operational but unfortunately, some of its important provisions have been too amended gradually.

### 3.3. Conclusive Picture of IA

Diagnosing the four symptoms of IA and assigning them to the seven forest policy areas described in the previous sections doesn't help too much unless finding solutions to the problems brought about by the forest restitution.

In Table 4 we tried to assess how important are the main causes of IA, to indicate the institutions able to address the problems and to formulate possible solutions, based on the experience gained in the near past or common-sense inferences. However, these solutions are not simple at all and institutional unsteadiness seems to be the chronic disease of the forestry sector.

**Table 4.** Summary of the main causes of institutional amnesia (IA) and possible remedies.

| Causes of IA | Severity of Symptoms | Who Shall Intervene | What Must Be Changed | Likelihood to Change in the Near Future |
|---|---|---|---|---|
| Resuming forestland restitution | High | Ministry of water and forests, Constitutional Court | New bills on restoration withdrawn or rejected by the Constitutional Court | High, |
| Delaying restitution process | Low | National Agency of Cadaster and Real Estate Advertising | Financial support for general cadaster, including forests, hunting cottages, forest roads | Quite high |
| Forest management planning | High | Ministry of Water and Forest | Integrated system of forest management (all applications, including forest planning) on a single platform) | Medium |

<div align="center">**Table 4.** *Cont.*</div>

| Causes of IA | Severity of Symptoms | Who Shall Intervene | What Must be Changed | Likelihood to Change in the Near Future |
|---|---|---|---|---|
| Institutional unsteadiness | High | All political parties | Political culture | Low |
| Horizontally harmonized legislation | Medium | Ministries of forests, agriculture, and environment | Less ministerial orders but better harmonized | Medium |
| Political disputes | Low | Parliament—commission of Agriculture and forestry | Professional structure of parliamentary commissions (Senate and Secondary chamber) | Low |

## 4. Discussion

By far the main cause of IA of forestry is the institutional instability, inherited from the communist period. In a command and control type of forest economy, based only on public ownership, the institutional memory was secured by the forest management plan, which sufficed. Whatever changes had occurred in the institutional settings, the forest management plans conveyed the same information to all stakeholders, able to make operational decisions in a consistent way.

Nowadays, insufficient horizontally harmonized legislation is another cause of IA but a great deal of the institutional memory can be retrieved from experience gained after 2007, when Romania joined European Union [58].

After the year of 2000 when the process of forestland restitution gained speed, sawmills mushroomed [59–61], and the central and regional authorities (i.e., FG and even NFA subsidiaries) lost their control on illegal logging, for different reasons, the most important ones being the institutional changes and poor salaries [14,18,62]. Better political culture needed to cure institutional unsteadiness is the hardest thing to attain, chiefly because most of the politicians do not understand the importance of having consistent public policies across more than two elective cycles.

Owing to the numerous trials prosecuted by NFA, attempting to deter the unfolding of the restitution process, the cutting budgets, as provided by forest plans, were no longer pursued, meaning that large amounts of timber were harvested wherever was possible and profitable, allegedly as salvage cuttings [50], in defiance to the traditional forestry rules applied before 1990. These wrongdoings fed mistrust in the utility of the forest planning system, especially among private forest owners.

The cluttered situations that had developed before (no cuttings at all), and after having the management plans of Natura 2000 sites (no cuttings, despite the provision of the forest management plans) also amplified the sense of getting lost in the bureaucratic procedures brought by the new legislation on biodiversity conservation, deemed as a black-box in this study. Not surprisingly, foresters have been complaining of being misunderstood by politicians [63] and mistrusted by environmentalists [64]. However, they ignore that political parties select their candidates according to the representational needs of the voters [65], which has little or nothing to do with forestry.

In a full democracy, a weak representation in the Parliament of professional foresters (18 deputies and senators in almost 30 years) could be another cause of promoting forest-related laws with intended or unintended hidden loopholes, since all Parliament members are influenced by the local stakeholders interested in maintaining whatever status quo, at the best [66,67]. And the most important loopholes of the forestland restitution laws were presented in Table 2.

Up to now the core principle of forest restitution was the historical justice, meaning that woodlands were given back to the ones from whom the communist regime confiscated the forests, or to their direct descents. A series of private joint ownerships simply didn't leave behind enough legal proofs of land tenure, and these forests remained in public property. To resume restituting these forests has nothing to do with social justice as long as the new claimants are not the endowed descendants of the

initial owners; it is no more land tenure restitution, it is privatization or re-privatization, and this may turn the "historical justice into a bureaucratic nightmare" [68]. Barely in 2019 the results of a survey deployed in 2016 in all nine FGs have been published and, not surprisingly, 73% of the FG workforce was complaining of unsuited training and 66% of unsuited legislation [69].

## 5. Conclusions

Beyond the historical equity dimension of restoring the private ownership of land and forests, all political parties that ruled Romania in the last three decades used the forest restitution as electoral baits, each new law promising more than the previous ones did. Therefore, important components of the forest policy (forest watching, illegal logging preventive measures, timber cruising, and forest planning) have always been reactive, not proactive.

Resuming the forest restitution process keeps on washing out the institutional memory of the forestry sector, whose weaknesses were inherited from the communist period. The new bureaucracy brought by the social and institutional networks wherein forestry is connected with other sectors comes with new information but less knowledge retrieved from the past via the forest planning system.

However, the tendency to overregulate the forestry sector just for preventing illegal logging didn't help too much because the Forest Code is an organic law and cannot be updated at the same pace with the laws operating in biodiversity conservation and other sectors. Given the special character of the environmental legislation, acknowledged not only at national level, but internationally too, the forest legislation should be subordinated to, and correlated with this legislation.

Forestry practices and environment protection should not be led in opposite directions; on the contrary, a sustainable forest management is based on, and is bounded by the legislation of environment protection. Mapping the four symptoms of institutional amnesia on the storyline of the last three decades of forestry in Romania is a necessary endeavor for identifying and understanding the roots of the numerous failures of the forest policy reported in literature and highlighted by numerous stakeholders.

On the long-term, the most important means to reduce the pace of IA is institutional steadiness but it depends on how the politicians understand that forest policy tends to be rather a public policy than a sectoral one.

**Author Contributions:** D.M., Conceptualization, methodology, and surveys. V.T., Legal framework analysis.

**Funding:** This research received no external funding.

**Acknowledgments:** We are deeply grateful to all professional foresters and interviewees who willingly shared with us their experience in handling so many laws and regulations. Without their support probably we would have stopped digging into this intriguing and controversial legal matter. We also acknowledge our anonymous colleagues of Transylvania University, who have scanned and put online the entire collection of Revista Pădurilor (continuously published since 1886), which is *de facto* the only institutional memory of Romanian forestry. Last but not the least, we are grateful to the three anonymous peer reviewers for their supportive feedback and suggestions.

**Conflicts of Interest:** The authors declare no conflict of interest.

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
