# Peer review of "Did Forestland Restitution Facilitate Institutional Amnesia? Some Evidence from Romanian Forest Policy"

_land, doi:10.3390/land8060099_

Round 1
Reviewer 1 Report
Review of the paper "Did forestland restitution facilitate institutional amnesia? Some evidence from Romanian forest policy"
The paper "Did forestland restitution facilitate institutional amnesia? Some evidence from Romanian forest policy" aims at identifying the institutional amnesia following the process of forest restitution occurring in Romania. First, I found the paper well written, with many sentences showing a rich vocabulary and easiness to form challenging ideas. However, as with any paper, improvements ae possible, and I have made several comments that could help in streamlining the discourse.
Secondly, I found the paper of significant importance in explaining the intricate evolution between technical personnel, political decision, and corruption. Few people were expecting that one of the results would be the loss of institutional memory, with all the disadvantages and advantages. Furthermore, the documentation in such details of the entire process that lead to institutional amnesia would serve, without a doubt, as a reference point in describing and modeling the impact of politics on forest and forestry.
In conclusion, I found the manuscript suitable for publication, pending the comments that I have made. To ensure that the authors would benefit from my recommendations I have added the pdf file which which I have inserted mu comments.

Author Response
Document reviewed according to all observation made by all three reviewers

Reviewer 2 Report
* The abstract should be rewritten. A good abstract should start with a background, followed by motivations, then the proposal of this research, then the adopted methodology, and finally the results. The current version of the abstract does not have such a structure, so that it is hard for readers to follow the results of this work based on the abstract.
* The paper title is related with forestland. However, the abstract and the paper content do not focus on forestland. If the paper title is not changed, then more content on forestland should be added.
* This paper focuses on investigating the Romanian case. Some parts in this paper sometimes suppose that the authors are familiar with the Romanian situation. However, this is not true. Hence, the authors should rewrite the whole paper carefully, to make sure that all readers (not just Romanian ones) can understand each word of this paper.
* The content of the Introduction section is too short. The related works should be reviewed, so that the motivations behind this work should be determined based on the previous works. In addition, if this work is related to the forestland, then the related works and the related Romanian situation on forestland should be introduced.
* In the Introduction, more details on the institutional amnesia should be given.
* In Lines 67 - 75: The authors use the concept of software development as an analogy to explain the problem. It looks unnecessary and not appropriate, because not all readers (especially, those that are not familiar with programming) can understand what the authors would like to express.
* In Section 2, the methodology adopted for analysis in this research should be explained.
* In Section 3, the map visualizations before and after the institutional amnesia (IA) should be provided, so that the readers can understand the degree to which the IA affects the land restitution.
* The results is based on five semi-structured interviews. All possible bias should be explained.
* In Section 4, the discussion is too simple. Based on the results analyzed in this paper, the authors should provide more insights, policy implications and suggestions, and potential improvement strategies.
* A large number of English typos and grammar faults exist in this paper. The authors should carefully check the whole article. Some but not all comments for English faults are as follows:
-- Line 9: `A The' --> `This'
-- Line 9: `restitution, that' --> `restitution, which'
-- Line 44: `have been' --> `has been'
-- Line 50: `This research note' ?
-- Line 52: `, that highlighted' --> `, which highlighted'
-- Line 54: `evaluating' --> `evaluate'
-- Line 65: `needs society' --> `needs that society'
-- Line 66: `the ones who' --> `the ones that'
...
Author Response
Comments and Suggestions for Authors
* The abstract should be rewritten. A good abstract should start with a background, followed by motivations, then the proposal of this research, then the adopted methodology, and finally the results. The current version of the abstract does not have such a structure, so that it is hard for readers to follow the results of this work based on the abstract. Completely changed
* The paper title is related with forestland. However, the abstract and the paper content do not focus on forestland. If the paper title is not changed, then more content on forestland should be added. I changed all over land with forestland though the three laws refer to land in general.
* This paper focuses on investigating the Romanian case. Some parts in this paper sometimes suppose that the authors are familiar with the Romanian situation. However, this is not true. Hence, the authors should rewrite the whole paper carefully, to make sure that all readers (not just Romanian ones) can understand each word of this paper. Partially done. We got rid of redundant or unclear paragraphs, simplified the complicated ones.
* The content of the Introduction section is too short. The related works should be reviewed, so that the motivations behind this work should be determined based on the previous works. In addition, if this work is related to the forestland, then the related works and the related Romanian situation on forestland should be introduced. We specified wherever was necessary forestland, not simple land restitution and we also added some more literature.
* In the Introduction, more details on the institutional amnesia should be given. Solved.
* In Lines 67 - 75: The authors use the concept of software development as an analogy to explain the problem. It looks unnecessary and not appropriate, because not all readers (especially, those that are not familiar with programming) can understand what the authors would like to express. Deleted.
* In Section 2, the methodology adopted for analysis in this research should be explained. We introduced policy-arrangement approach as methodological core and we re-arranged the results according to this methodology
* In Section 3, the map visualizations before and after the institutional amnesia (IA) should be provided, so that the readers can understand the degree to which the IA affects the land restitution. We wrote a new sub-section of results and table 4.
* The results is based on five semi-structured interviews. All possible bias should be explained. The interviews were meant to check whether all important side-effects of restitution laws have been addressed and not to collect more opinions about the process itself.
* In Section 4, the discussion is too simple. Based on the results analyzed in this paper, the authors should provide more insights, policy implications and suggestions, and potential improvement strategies.
Discussion developed
* A large number of English typos and grammar faults exist in this paper. The authors should carefully check the whole article. Some but not all comments for English faults are as follows:
Fixed
-- Line 9: `A The' --> `This' solved
-- Line 9: `restitution, that' --> `restitution, which' solved
-- Line 44: `have been' --> `has been' solved
-- Line 50: `This research note' ? the article was intend as commentary.
-- Line 52: `, that highlighted' --> `, which highlighted' solved

Reviewer 3 Report
Dear Authors,
Although the material refers to scientific papers and supports some parts with these references, the material is rather descriptive, objective research methods are not applied or decsribed sufficiently, thus I consider it rather suitable for newpapers than for a scientific journal. Broader and better ballanced team of authors, including Romanian state forest administration would contribute to the quality of paper on the forest policy, restitution and management ruling.
Some more detailed comments/suggestions:
25-61 Introduction should provide more insight on motivation for this work.
229-237 Citing statements of a single person as owner (of thousands) is not objective, rather a summary of a larger survey would be appropriate.
253 Please check correctness of reference to the Figure 3 / Figure 2.
265 Please check correctness of reference to the Figure 2 / Figure 1.
274 -281 Fig. 3 An explanation of terms 1st/2nd appeal, annulment, trial court would be appropriate for better understanding.
484-497 Discussion has not character of discussion on the topic of the text above.
498-524 Conclusions rather provide extra comments on the context than substance of results.
Author Response
Although the material refers to scientific papers and supports some parts with these references, the material is rather descriptive, objective research methods are not applied or decsribed sufficiently, thus I consider it rather suitable for newpapers than for a scientific journal. Broader and better ballanced team of authors, including Romanian state forest administration would contribute to the quality of paper on the forest policy, restitution and management ruling. All interviewees but one (the forest ower) worked with NFA. The main author was member of the Administrative council of NFA.
Yes. The article is more descriptive than analytic and it falls into „commentary” category. The institutional amnesia has been perceived rather by public authority than National Forest Administration.
Some more detailed comments/suggestions:
25-61 Introduction should provide more insight on motivation for this work. Addressed
229-237 Citing statements of a single person as owner (of thousands) is not objective, rather a summary of a larger survey would be appropriate. We opt for a single big forest owner because he went through all trials with NFA, it was explained in the text.
253 Please check correctness of reference to the Figure 3 / Figure 2.Solved
265 Please check correctness of reference to the Figure 2 / Figure 1. Solved
274 -281 Fig. 3 An explanation of terms 1st/2nd appeal, annulment, trial court would be appropriate for better understanding.(Solved, explanation is in methodology section)
484-497 Discussion has not character of discussion on the topic of the text above. Changed.
498-524 Conclusions rather provide extra comments on the context than substance of results. Reformulated and shortened

Round 2
Reviewer 2 Report
We have revised the manuscript as suggested and the English can be polished.
Author Response
I revised the English and simplified a little bit the text. I also added two additional references (www links): the European Court of Human Rights and the link to Revista Padurilor, the Romanian oldest professional journal, which helped us to compile the history of the institutional settings since the end of the WWII. I also moved two paragraphs in order to better fit the text into the page, and make a more coherent reference to (forest plans) addenda” (see lines 303, 399). I also replaced a lot of words with appropriate English synonyms (like body instead of organism)...

Reviewer 3 Report
Dear Authors,
Thank you for updated manuscript.
While the information was extended where recommended, there are still some minor issues which may be improved, e.g. the sentence at line 78 seems to be not completed, Figure 2 does not contain state forests category in the legend. Please check the text thoroughly before re-submission.
Provided manuscript has been significantly updated, considering my previous comments.
However, the document still lacks character of a scientific paper, and you argue in your response that "it falls into „commentary” category." Indeed, provided material would fit such a category.
Author Response

(The authors gave the same response as above.)
